# Socioeconomic effects of water hyacinth (*Echhornia Crassipes*) in Lake Tana, North Western Ethiopia

**Belachew Getnet Enyew** [1]*, **Workiyie Worie Assefa** [2], **Ayenew Gezie** [3]

**1** Blue Nile Water Institute and Faculty of Social Sciences, Bahir Dar University, Bahir Dar, Ethiopia, **2** Blue Nile Water Institute and Department of Biology, Bahir Dar University, Bahir Dar, Ethiopia, **3** Department of Biology, College of Science, Bahir Dar University, Bahir Dar, Ethiopia

* belachew_getnet@yahoo.com

**Data Availability Statement:** All relevant data are within the manuscript and its Supporting Information files.

**Funding:** The authors received no specific funding for this work.

## Abstract

Water hyacinth has been progressively advanced in Lake Tana since 2011 and covered vast areas of the lakeshore. The aim of this study was to assess how the lakeshore covered by the weed mats affected the socioeconomic of the local community. The study was based on a survey of 405 households, 8 group discussions and interviews of 15 key informants conducted from January to March 2018. The results revealed that crop production, livestock feed supply, water supply, fishing, the health of local people and livestock were impacted negatively by the infestation of water hyacinth. The range of socioeconomic problems caused by the weed generally implied the real impacts on the lives of local communities and national economic development. The efforts made to control water hyacinth has costed huge labor and financial resources. The results revealed that close to 800,000 human labor dedicated to manual removal of the weed from 2012 to 2018 and above one million USD spent for procurement of harvester machines and bioagent experiments. In spite of the devotion of huge labor and spending of a lot of money, the expansion of the weed has not controlled. Poor coordination of controlling efforts, dumping of harvested dense mats of the weed in the lakeshore, lack of genuine participation of the local people are principal factors for the failure of the controlling efforts A coordination of various stakeholders thus is needed to make eradicating methods more effective. Other alternative options should also be considered to control the weed expansion.

## Introduction

Invasive alien plant species (IAPS) are extensively expanded globally in the twenty-first century [1]. One-sixth of the global land surface is highly vulnerable to IAPS invasion including substantial areas of developing countries and biodiversity hotspots [1]. Because of the fertile ground created by high levels of nutrient loading, the extent of IAPS is the most serious in inland fresh water ecosystem [2]. An extensive expansion of IAPS has threatened the global biodiversity, economies and human health [1,2]. Water hyacinth (*Eichhornia crassipes*) is one

**Competing interests:** The authors have declared that no competing interests exist.

of the world's most prevalent invasive aquatic plants. It is a free-floating plant well known for its reproduction abilities because of its rapid growth rate [3]. Even though water hyacinth is a native of Brazil and other South American countries, it has invaded the lakes and swamps in most countries of the world lying between 40°N and 40° S [4]. Water hyacinth was officially reported in Ethiopia about 60 years ago in Koka Lake and the Awash River [5]. However, the area infested at that time was small. The weed had spread into Blue Nile, Baro-Akobo and Rift Valley Basin Systems in the 1980s and 1990s [5]. The exact time of water hyacinth infestation in the Lake Tana shore is not well known. Stroud [5] reported that the plant was observed in marshy tracts of the Abay River near its outlet with Lake Tana in early 1990s. The preliminary survey done by [6], however reported that the weed was observed in Megech river mouth for the first in 2010. Officially, water hyacinth was recognized as ecologically dangerous invasive weed in 2011 [4].

The total area of Lake Tana shore infested by water hyacinth varied from year to year. [7] estimated the surface area of the lakeshores covered by the mats ranged from 200 sq. km (2012/2013 to 500 sq.km (2014/15). Environmental, forest and wildlife protection and development authority (EFWPDA), on the other hand, estimated the total area infested by the alien species ranged from 28.33 sq.km (2017/18) to 255.33 sq.km (2016/17). EFWPDA reported that the size of the surface area infested by the weed has been determined by the amount of annual rainfall. In line with this, [8] reported that heavy winds and waves associated with heavy rain significantly contribute to the variation in extent of water hyacinth infestation and mat size in the Lake Victoria basin.

Empirical studies have conducted on socioeconomic effects of water hyacinth. The studies reported that the weed has dichotomous socioeconomic impacts. Most of prior studies reported various negative socioeconomic impacts of the weed. One of the impacts of water hyacinth is its negative effect on paddy crop production. [9–11] reported that the plant affected paddy crop production by destroying the plant, inhibiting germination and interfering with harvesting. The weed has also affected fisheries and fish-related commercial activities [9], hindered water transportation [12,13] and decreased the quantity of water supply and reduced water quality [14]. Furthermore, the floating plant and its mats have led to worsening health conditions of the people by providing a breeding ground for mosquitoes, worms, snails, etc. [14], obstructed electricity generation [12,13] and blocked access to recreational areas [14].

Although water hyacinth is often seen as a weed responsible for various socioeconomic problems, it is also used for different benefits. Waste water treatment [15–17] and fertilizer for the nutrient deficient soils [18] are the most important benefits of water hyacinth. The plant has also used as fodder for livestock [19], renewable energy source [3] and rope, baskets, pulps and paper making [20]. However, the socio-economic benefit of water hyacinth is insignificant compared to the socio-ecological damages that it has incurred [3].

The benefits of water hyacinth reported by prior studies would not be expected in the Lake Tana since efforts have not been done yet by the scientific community, the local people and other stakeholders to change it into various benefits. The study done by [21] showed the negative effects of water hyacinth infestation in the Lake Tana shore on livestock, fishing and crop production. This paper, however, did not investigate the influence of the weed on livestock, fishing and crop production in detail. Besides, the paper has not assessed the effect of the weed on other resources that are basic to the livelihoods of the local people. As reported by [22] and [23], Lake Tana has provided important resources in addition to fishing, farmlands and livestock feeds. These resources consist of water for various uses, medicinal plants, craft making and building materials. The lake has also provided water transport, recreational and spiritual services. Furthermore, the lake significantly contributes to national economic growth through the generation of hydropower and the supply of irrigation water [22,23].

Evaluating the socioeconomic effect of water hyacinth is very difficult because of two important reasons. First, the full ranges of socioeconomic costs of biological invasions are often beyond the immediate impacts [24]. Second, the awareness of the local community on the consequence of invasive weeds on non-marketable services provided by the lake ecosystem is low. Cognizant of this fact, this study aimed to analyze the effect of water hyacinth that are immediately observed on crop production, fishing, water use for various uses, livestock feeds, human and livestock health, tourism and water transport services, and the cost of controlling water hyacinth.

## Description of the study area

Lake Tana, the head of the Blue Nile River, is located in the northwestern highlands of Ethiopia at an elevation of 1830 m. Geographically, it extends from 11˚ 0″ to 12˚ 40′ 0″ N latitude and from 36˚ 40′ 0″ to 38˚ 20′ 0″ E. longitude (Fig 1). The surface area of the lake is approximately 3,060 sq.km. It is a shallow lake with the maximum and average depth of 14 m and 6 m, respectively. The climate of Lake Tana is characterized by a major rainy season with heavy rains, during June-October, and sometimes a minor rainy season during February-March. The mean annual rainfall of the lake area is about 1280 mm and the mean annual temperature is 21.7˚C. The lake has more than 40 tributary rivers, of which Gilgel Abay from the south, Ribb and Gumara from the east and Megech River from the north are major rivers feeding the lake. The only river flowing out of Lake Tana is the Blue Nile River (Abay River). Lake Tana is endowed

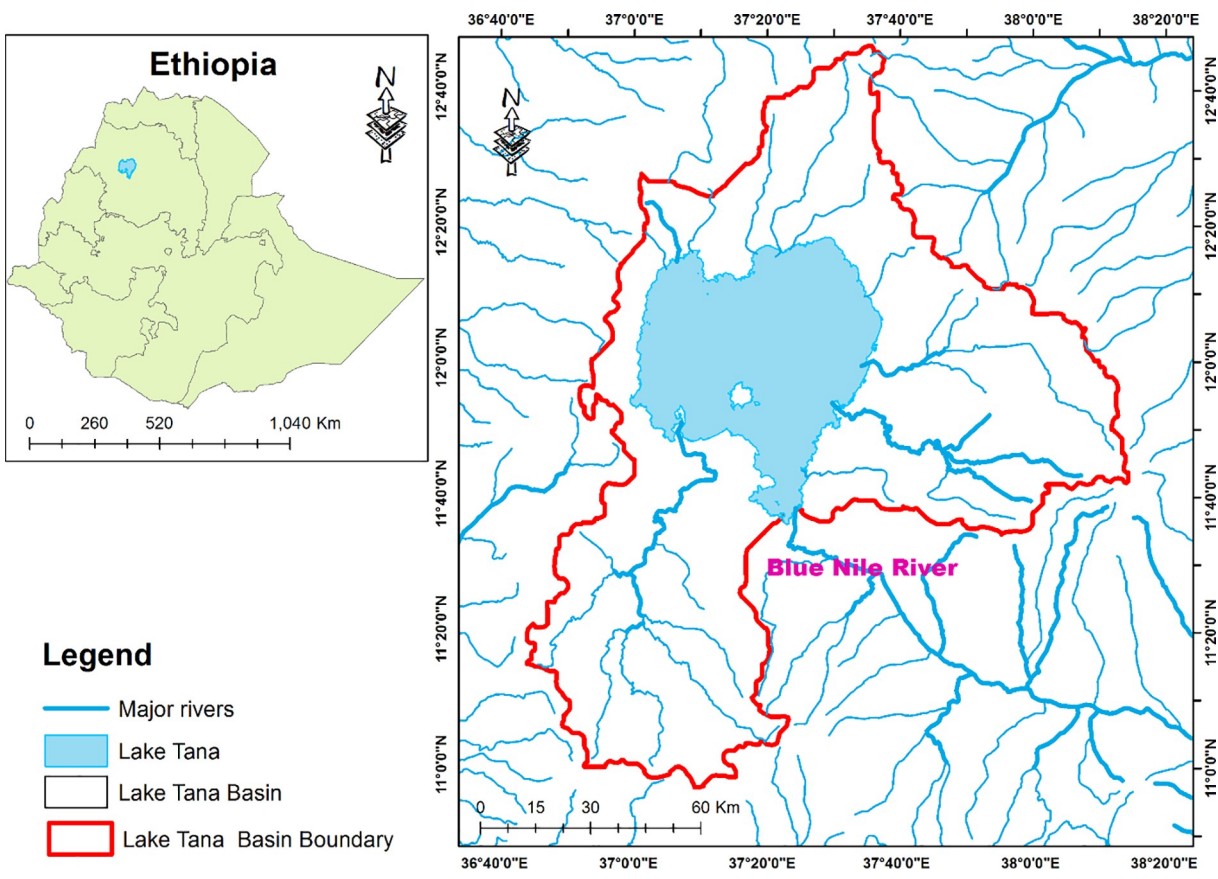

**Fig 1. Map of the study districts and sample *kebele* administrations.**

with a large number of wetlands that are among the largest and ecologically most important ones of Ethiopia. These wetlands harbor a rich flora and fauna, including endemic fish species of *Labeobarbus*, endemic amphibians, globally threatened waterfowl, large mammals such as *Hippopotamus amphibius* or critically endangered Cyprus Papyrus.

The lake region is one of the highly populated areas of Ethiopia [22]. The primary economic activity of most people is subsistence mixed crop-livestock farming system. Crop production is mainly rain-fed. Cereals such as '*teff*', maize, finger millet, rice, grass pea and chickpea are dominant crops. Livestock husbandry has also a significant contribution to the livelihood of farmers. It supports crop production mainly through providing traction power and providing transport services, and manure. They are also a source of cash income through the sale of the products and live animals.

## Research methods

### Ethical consideration

This research project proposal was evaluated by research evaluated committee established by Blue Nile Water Research Institute which is under Vice President of Research and Community Service of Bahir Dar University. The committee approved the research design and after the committee approval, the research proposal was presented in the workshop organized by Vice President of Research and Community Service of Bahir Dar University. Letter of requesting supports were written from the office of Vice President of Research and Community Service of Bahir Dar University to different zonal and woreda administration offices after Blue Nile Water Research Institute approved the research project. Then the permission letters were obtained from the zonal and woreda offices of agriculture, chief administrative office and office of environmental protection. Respondents were fully informed about the purpose of the research, and how the data obtained them would be managed and used and gave verbal consent. Sample households were coded by numbers rather than the family names. Confidentiality of information was assured by all the data collectors and investigators.

Informed consent was also obtained from key informants and the participants of FGDs before the interviews were conducted. Both KII and FGDs were conducted in places where key informants and the participants of FGDs feel that their confidentiality is completely protected. The places, where KII and FGDs took place, were selected based on key informants and FGDs participant suggestion.

### Sampling procedures

A multi-stage sampling approach was used to select households to be interviewed. The first stage in the sampling procedure involved selection of three districts such as Dembia, Gonder Zuria and Libo Kemkem purposely because these districts were infested by the weed. Four sample rural kebele administrations (KAs) from these three districts were selected purposely as well because the weed was observed for the first time in lake shore areas of these KAs and the area coverage of the weed was wider than the rest of infested KAs [7]. Four hundred eight rural households were selected from the sample KAs. The proportionate sampling technique was employed to determine the sample size of each sample KA. The total sample size of households was determined based on the following equation given by Israel (1967).

$$n_0 = \frac{N}{1 + N(e)^2}$$

Where = $n_0$ is sample size; N, is the total number of households. Based on the data obtained

from KAs agricultural office, the total number of households of selected KAs was 5120 during the survey period; and *e* is the level of precision (in this case 0.05). Considering the probability that some of the respondents might not available for interview or refuse to be interviewed, 10% of sample households were added. Sample households were systematically chosen by selecting every k$^{th}$ element of the population, where k is the sampling interval. The first number of the sample was selected randomly from within the first k elements. The lists of households in each of the KAs were obtained from the respective *kebele* leaders and agricultural extension agents.

## Methods of data collection

Secondary and primary data were collected using various techniques. Secondary data garnered from as the annual reports of EFWPDA, Bahir Dar City Tourism Development Office and Lake Tana Public Water Enterprise. The data collected from reports of these institutions included the number of people engaged in the weeding campaign, the area covered by water hyacinth and the size of crop fields and pasture lands infested by the weed, the effect of the weed on water transportation and hydroelectric power, etc. Moreover, the data about socioeconomic impacts of water hyacinth were collected from electronic and hard copy literature sources. Primary data were collected via household survey, Focus Group Discussions (FGDs) and Key Informant Interviews (KIIs).

## Household survey

A household survey (S1 Questionnaire) was undertaken from January to March 2018. The aim of the survey was to garner the data that helped to quantify the extent of water hyacinth effects on socio-economy of the local people in the infested areas in particular. The survey employed a structured questionnaire with pre-coded types of questions. The questionnaires focused on the following issues: (1) household demographic characteristics (age, sex, educational level, family size and marital status); (2) ownership and use of land; (3) livestock holding and sources of animal feed; (4) the rate of water hyacinth infestation (5) negative effect of water hyacinth (crop production, livestock feed, fishing, water supply, animal and human health, wetland resource utilization); and (6) controlling mechanisms of water hyacinth and their pitfalls. The retrospective-focused (before-and after infestation) question items were included within cross-sectional survey questionnaire to capture the changes due to water hyacinth invasion in Lake Tana shores. A pretest interview was held with 30 farmers in four sampled KAs prior to the actual data collection enabling to polish the questionnaire. The survey was administered by 16 trained data collectors after pre-testing of the questionnaire. The questionnaires were completed by visiting households around their homestead. The survey work of the data collectors was crosschecked by re-interviewing 39.2% of the households surveyed by data collectors to confirm the reliability of the collected data.

## Key informant interview

Key informant interviews were conducted with 15 persons including experts and heads of *woreda* agriculture and environmental protection offices, chief administrators of KA, experts of environmental, forest and wildlife protection and development authority (EFWPDA) and Lake Tana Public Water Enterprise. These interviews were semi-structured and mainly focused on the trends of water hyacinth infestation, the effects of the weed on the socio-economy of local people, water hyacinth controlling methods and their roles in controlling the infestation rate of the weed, etc. The interviews were recorded by audio-records with consents of the key informants.

### Focus Group Discussions (FGDs)

A total of eight FGDs were conducted, two per sample KA to check and refine the data collected from household surveys and KIIs. The first FGD was held with persons consisting of both men and women, elders, youth and prominent individuals and the second FGD was held with fishermen. Each group had a membership range between eight and ten. The identification of the participants of FGDs was done in consultation with agricultural extension agents and chief administrators of KAs assuming that they are helpful in identifying individuals who are knowledgeable and thought to have particular insights and opinions about the subject under investigation. Open-ended interview guild was developed for the FGDs The key issues of the discussions were the magnitude of water hyacinth infestation in the lakeshore and associated wetlands, the effect of the weed on crop production, animal feed supply, fisheries, water supply, animal and human health; community participation in the controlling of the weed, the effect of community's engagement in controlling water hyacinth, etc. Sound records were used to record the group discussions with participants' consent.

### Methods of data analysis

The data generated by the structured questionnaire were analyzed using descriptive statistics such as percentage, frequency tables and mean. In addition, analysis of variance (ANOVA) was used to compare differences regarding labor costs of farming activities (ploughing, weeding, harvesting) before and after water hyacinth infestation, as well as the cost of fishing equipment and amount of fish catch before and after the weed infestation. A diagnostics test was done before ANOVA to check whether or not the data was normally distributed using Shapiro-Wilk Test. The values of Shapiro-Wilk Test for both variables were found to be above 0.05. This indicates that data is not deviated from a normal distribution. Homoscedasticity test was also done and the value of the test found to be above 0.05. This showed the homogeny of the data variance. Data analysis was carried out using Microsoft excel and statistical package for social sciences (SPSS) software version 20. The audio-records that were taken during FGDs and KIIs were first transcribed into English and thoroughly read and finally, transcripts were analyzed using thematic analysis.

## Results and discussion

### Key demographic and socioeconomic characteristics of the respondents

A total of 405 households were responded a questionnaire. More than 90% of them were headed by males. The average age of household heads was 45.4 years, but ranged from 19 to 85 years. Nearly 65% respondents were under the age group of 22–50 years. They had an average family size of 5.0 persons, which was relatively higher compared to the national average (4.7 persons) [25,26]. The majority of households were illiterate (46.8%), nearly 1/3rd was able to read and write only, and the remainder had some level of formal education. Close to 90% of them kept different types of livestock. They owned on average 3.63 tropical livestock units (TLU) varying from 0.02 to 21.3 TLU. As it is other rural parts of Ethiopia, land is the most important livelihood asset for close to 80% households. The average land holding was 1.53 hectares per household, with a range from 0.25 to 4.25 hectares. This is higher than the national average of 1.14 hectare per household [26]. Nearly 40% of respondents owned less than 1 hectare. These households were engaged in different livelihood activities. On-farm activities remained the principal sources of employment despite the declining of farmland per household. A quarter of households were also found to be engaged in off and non-farm activities such as wage labor and petty trading.

## Socioeconomic impacts of water hyacinth

All of the respondents were aware of the presence of water hyacinth in Lake Tana shore. Close to three-fourth of the respondents observed the infestation of the lakeshore and the mouth of its tributaries by the weed seven years ago and the rest reported that water hyacinth invaded the lakeshore in the past 5 years. More than half of the respondents did not know in what way water hyacinth come to Lake Tana, whereas 37.7% and 8.9% of the respondents thought that the weed came to Lake Tana by visitors and migratory birds, respectively. When the respondents were asked to describe the impact of water hyacinth, either positive or negative, all of them stated the negative impact. The results of the survey and qualitative data reported that crop production, fishing, livestock feed, water supply, water transportation and other economic activities were affected by water hyacinth. The findings of both the survey and qualitative data are reported and discussed as follows.

## Impact on crop production

Crop production is the main source of food and means of income generation of the people living in the Lake Tana area. The most commonly grown crops were *teff*, rice, finger millet, grass pea and chickpeas. More than 80% of the farm land was allocated for the cultivation of these crops. Rice, vetch and chickpea took the lion share of cultivated land (64%) and they are grown in the marshy lands found near to the lake shore, Rice is grown during wet season while chickpea and grass pea are cultivated during dry season using either moisture or irrigation by using the lake water.

The infestation of water hyacinth in Lake Tana has affected crop production. More than 70% of interviewed farmers' crop fields were infested by the weed in 2014/15 and 2016/17 cropping seasons. It is similar with the findings of [27]. According to [27], the crops production of close to 70% of farmers was affected by water hyacinth in the Rift Valley of Ethiopia. Farmers were asked to level the intensity of water hyacinth infestation on their crop fields. As presented in Table 1, the infestation levels were 'severe' by 47.8% of respondents, 'moderate' by 24.4% respondents and the remaining 27.8% respondents rated it as scattered. Based on the survey results, about 43.4% (2 sq.km) of crop fields were vulnerable to water hyacinth infestation during the peak seasons of the weed. Of which, 43.7% of infested area was severely affected, a quarter of infested area was moderately affected and the rate of infestation in the remaining cultivated area (30.4%) was sparse (low).

The loss of crop production varied across the plots depended on the weed infestation rate. In severely affected crop fields, about 75 to 100% of crop production was lost. The loss of crop production in moderately infested crop fields was also ranged from a quarter to one-third of production during the peak infestation seasons. Based on the survey results, the rice grown in 0.9 sq.km of cultivated land in 2014/15 and 2016/17 was totally damaged by the dense mats of water hyacinth that was transported by the lake wave. On the average, about 30% of rice that could be produced from 0.54 hectares of land was also lost due to the weed infestation.

**Table 1. The extent of water hyacinth infestation in the crop fields of interviewed households.**

| Rate of infestation | Households | | Infested Land | | Estimated crop loss (%) |
|---|---|---|---|---|---|
| | No | % | Area (sq.km) | % | |
| Sever | 139 | 47.8 | 0.9 | 43.7 | 75–100 |
| Moderate | 71 | 24.4 | 0.54 | 25.9 | 25–33.3 |
| Scattered | 81 | 27.8 | 0.63 | 30.4 | 0–10 |
| Total | 291 | 100 | 2.07 | 100 | |

Estimation made based on the yield of rice (3 tons/ha) harvested during good harvesting season, the farmers lost close to 305 tons of rice. As estimated based on the farmgate price of rice, the total amount of rice production damaged by the weed was estimated to be 149,226 USD. The interviewed farmers, whose rice fields damaged by the weed's dense mats in 2014/15 and 2016/17 cropping seasons, also reported that they did not grow chickpea, grass pea or vegetable crops in 0.3 sq.km of land due to difficulty of clearing the mats from rice fields. The total amount of chickpea and grass pea production lost was estimated to be 40 tons which was equivalent to USD 21,333.

The substantial loss of crop production was also corroborated by the *woreda* agricultural offices. According to *woreda* offices of agriculture, about 52 sq.km of cultivated land in 21 KAs was vulnerable to water hyacinth infestation in 2014/15. Of which, 15.5 sq.km of rice fields were totally damaged by the weed during cropping season of 2014/15. Similar size of farmland grown by rice was also affected by the dense mats of the weed during 2016/17 and 2018/19 cropping seasons. The total amount of crops that could be produced from the damaged rice fields was estimated to be 446 tons. In terms of money, it was estimated to be USD 2, 179,584.

Creating the work burden on farming activities was also another effect of water hyacinth on crop production. The findings of FGDs indicated that the weed has affected agricultural activities such as ploughing, weeding and harvesting. During the discussions, the participants reported that the dense mats of the weed piled in crop fields during weeding campaign have made ploughing very difficult. Avoiding the heap of harvested mats needed extra labor during ploughing. As presented in Table 2, the labor that needed to plough a hectare of land infested by water hyacinth, was 73.6% higher than the labor needed to plough a hectare of land before infestation. In monetary terms, extra USD 19.40 was required to avoid the harvested mats of water hyacinth during ploughing. ANOVA revealed that the difference between the number of labor/ha and estimated cost of labor was significant at $P < 0.01$ and 0.05, respectively. Similarly, the labor that was needed for weeding and harvesting of the plot infested by the weed was higher by 85.4% and 34.9% than non-infested plot, respectively. In terms of cost, a farmer spent more money for weeding and harvesting of crops in a hectare of farmland than twice that of non-infested farmland. The difference was is statistically significant at $p<0.001$ as confirmed by analysis of mean variance.

The effect of water hyacinth on the farming activities was also observed along River Tano and Tano Lagoon, Ghana [28]. The influence of the weed on farm activities of smallholder farmers however was not creating the work burden by increasing labor demand like Lake Tana area. According to [28], the weed affected the farming activities by making the crossing of the river very difficult and creating a challenge to the proper management of farm fields.

**Table 2. The effect of water hyacinth on the cereal crop farming activities.**

| Agricultural Activities | Labor/ha | | F-Value | Estimated Cost (USD) | | F-Value |
|---|---|---|---|---|---|---|
| | Before | After | | Before | After | |
| Ploughing | 9.5 | 16.41 | 47.336*** | 26.54 | 45.92 | 21.6** |
| Weeding | 24 | 44.49 | 15.82*** | 34.37 | 70.60 | 4.48** |
| Harvesting | 17.25 | 23.27 | 286.35** | 18.57 | 41.60 | 3.289** |

Source: Computed from survey Data.

Note: * = significant at p<0.1

** = significant at p<0.05

*** = significant at p<0.001

## Impact on fishing

Fishing and associated businesses are a way of life for many communities living around Lake Tana. According to [26], fishing was the livelihood activity of 40,000 fishers, fish processors, traders, gillnets makers, boat builders, etc. The data obtained from woreda agricultural offices, about 2, 905 people were engaged in commercial fishing in 21 infested KAs. Fish resources of Lake Tana are subjected to the severe decline. Fishing pressure during the peak spawning season, using illegal gillnets, overfishing, degradation of breeding and feeding grounds of the fish and water pollution are the factors that responsible for the declining of fish stock [29]. The invasion of lakeshore, river mouths and wetlands by water hyacinth has exacerbated the deterioration of fish population of Lake Tana. During the discussions, fishermen described that the infestation of lakeshore, river mouths and swamps by water hyacinth has made the problem በእንቅርት ላይ ጆሮ ገድፍ literally mean mumps on a goiter.

The effect of water hyacinth on different fish species is not the same. The participants of fishermen FGDs observed that O*reochromis niloticus* and *Labeobarbus species* are the most seriously affected species because of obliteration of their spawning areas and entangled by water hyacinth. Similar result was reported in the Wouri River Basin, Cameroon by [30]. [30] observed the decline of eight fish species. O*reochromis niloticus* and barbus species were among the affected species by the weed infestation. Contrary to the findings of [30], catch fish was not influenced by the weed in Lake Tana. During the discussions, fishermen reported that cat fish was abundant in the lake shores and wetlands where the dense mats of water hyacinth were covered. Further study is needed to investigate the reason why cat fish is not affected by the weed. The fishermen however thought that the capacity of this fish species to penetrate the dense mats of the weed and relative availability of its feed are the principal factors for less influence water hyacinth on cat fish.

The participants of fishermen discussions stated that the quantity of fish they caught per day after water hyacinth expansion was lower than pre-water hyacinth expansion. The survey data similarly showed the substantial difference between the number of fish caught in pre-and post-water hyacinth infestation. Based on the survey results, on average, an individual fisherman could catch 28 kilograms of fish /day during spawning season before water hyacinth infestation. The average weight of fish caught after the infestation of the weed was decreased by 46.4% (13 kilograms/ day) at the same period. Statistically significant difference ($F$ = 6.12; $P$ <0.01) was noted between the average weight of fish caught before and after the weed infestation. The reduction of the quantity of fish caught per fisherman after water hyacinth infestation was also reported in previous studies [30–32]. [30] reported that the mean daily fish catch per fisherman had decreased by 90% after the infestation of the Wouri River Basin, Cameroon by water hyacinth compared with pre-infestation. The findings of fishermen FGDs indicated that the elimination of fish feed and obstruction of fish migration to the spawning habitat by the weed were accounted for the reduction of the amount fish caught per fisherman. In related studies [31–33], it was reported that the decline of fish population because of the destruction of its breeding ground and the blockage of fishing ground accessibility by the weed dense mats are caused for the decline of fish catchability.

The quantity of fish catch reduction resulted in a corresponding decrease of the income of fishermen. Estimation made based on the present value of (PV) of 2010, about USD 49.80 lost per fisher in a single day during spawning season. There was a significance difference ($F$ = 5.04; $P$ < 0.01) between the estimated income generated from fish sale before and after the weed infestation. A decrease of fishermen income was also observed by [30] and [32] along the River Tano and Abby Tano Lagoon, Ghana. [30] reported that the monthly income fishermen were decreased by three-fourth after the weed infestation compared with before invasion.

Increasing the cost of fishing was another effect of water hyacinth infestation in Lake Tana as it was observed in Lake Victoria [3]. The findings of fishermen discussions reported that water hyacinth has increased the annual cost of fishing by shortening the lifespan of reed boats and gillnets. During the discussions, fishermen reported that the gillnets were served from seven to nine months before the lake, swamps and river mouths were invaded by water hyacinth and a traditional boat made from papyrus reeds was served us 8 to 9 months. However, the weed has shorted the lifespan of gillnets and reed boats services to 3–4 months by damaging and transporting into other areas.

The survey results also testified the shortening of the lifespan of the reed boats and gillnets and its effect on increasing fishing cost. Estimation done based on the present value of (PV) of 2010 revealed that each fisherman had spent USD117 per annum after the weed infested the lakeshore. This was higher by 80.3% of the gillnet cost before the weed infestation. The analysis of mean variance (ANOVA) indicated that difference between the cost of gillnets before and after infestation was significant at $P < 0.001$. Similarly, the cost of traditional boats spent after the weed infestation (USD 11.95) was twice higher than the corresponding figure of pre-water hyacinth infestation. Statistically significant difference ($F = 6.14$; $P < 0.001$) was also observed between the average cost of traditional boats before and after the weed infestation. These findings are consistent with the results reported in other countries of fresh waters [30,31]. During the discussions, fishermen participants reported that the low fish catching per gillnet and increasing cost of fishing equipment have made fishermen hopeless and looked for other livelihood options.

## Impact on livestock feeds

Farmers in the study areas raised livestock for various purposes. The dominant types of livestock reared included cattle, sheep, donkeys and poultry. The livestock holding per household was relatively larger as compared to other areas [34,35]. As presented in the Table 3, the holding size was found to be 3.84 TLU. This figure was higher than the corresponding figure at national level (3.63 TLU) (CSA, 2018). The ownership pattern of livestock showed that about 69.7% livestock holders owned from 0.05 to 5.0 TLU, a quarter of holders (24.4%) owned from 5.01 to 10.0 TLU and about 5.8% owned above 10 TLU. Raising relatively larger number of livestock in the study areas as compared to others might be because of the availability of livestock feed and water in the lakeshore and its associated wetlands.

The livestock feed sources were grazing pasture, crop residues, crop aftermaths, hay, weeds and agro-byproducts. Amongst the feed sources, communal grazing pastures were the major one contributing large amounts of livestock feed for all herders. Based on the FGDs results, the lakeshores and wetlands were the major source of grazing pasture for people residing in KAs near to the lake. Based on the data obtained from *kebele* agricultural offices, the total number of cattle, sheep, goats and donkeys in 21 kebele administrations infested by water hyacinth was

**Table 3. The livestock holding size of respondent households.**

| Type of livestock | % of holders | Number | | | Tropical Livestock Unit | | |
|---|---|---|---|---|---|---|---|
| | | Min | Max | Mean | Min | Max | Mean |
| Cattle | 83.0 | 1,0 | 24.0 | 3.89 | 0.80 | 19.20 | 3.11 |
| Sheep | 32.6 | 1.0 | 23.0 | 1.49 | 0.10 | 2.30 | 0.15 |
| Goats | 6.4 | 1.0 | 6.0 | 0.23 | 0.10 | 0.60 | 0.02 |
| Donkey | 33.6 | 1.0 | 4.0 | 0.49 | 0.50 | 3.00 | 0.24 |
| Chicken | 42.5 | 2.0 | 30.0 | 2.92 | 0.02 | 0.30 | 0.03 |
| Total | 88.9 | | | | 0.05 | 33.70 | 3.84 |

74,750, 18, 965, 2,389 and 9,387, respectively. All livestock populations have freely set to graze in the lakeshore and surrounding wetlands. The grazing pasture of the lakeshore and associated wetlands have been degraded over time due to farmland expansion and overgrazing. Based on the survey results, all of interview farmers rated the grazing pastures of lakeshore and wetlands to be the first feed resources from November to June long years ago. The findings of FGDs also revealed that the lakeshore and wetlands were the sole sources of feed during dry seasons. As reported by FGDs, different plant species of grasses, sedges and broadleaves were grown in the lakeshore and marsh lands and thus the availability of pasture was adequate and able to accommodate large herds of cattle.

The lakeshore and wetlands' contribution in supplying livestock feed however has been declining. The illegal encroachment of farmland into the lakeshore and wetlands was accountable for the loss of grazing pastures. During discussions, the participants reported that large areas of the lakeshore and wetlands were converted to cultivated lands illegally in the past 30 or more years. The local government officials have also been allocated communal grazing lands for landless young households. The loss of grazing pastures in lakeshore and associated wetlands has been exacerbated by the invasion of water hyacinth. Due to the invasion of the weed, plant species have destroyed. Based on FGDs, the most important grass species that are destroyed by the massive mats of water hyacinth are *asindabo*, *gicha*, *denguha*, *kebegicha*, *telis*, *serdo*, *toka*, *ketema*, *selselo*, *yebahir ageda*, *shasho*, and *lelisa*.

In the study area, water hyacinth could not be an alternative to the damaged palatable grasses, sedges and broadleaves which is contradictory to the results reported by [18] and [36]. [18] claimed that water hyacinth could be used as a fodder for cows, goats, sheep and chickens due to its high protein content. During FGDs, the participants reported that the cattle are feeding water hyacinth before its flowing stage. The feeding of the weed, however, has affected the health of the cattle. Consequently, more than 90% of cattle holders prefer to confine their livestock in home compounds and used crop residues as a major source of feed. Based on the survey results, all of livestock raisers rated the crop residues as their first feed resources since the weed has infested the lakeshore and associated wetlands. The heavy reliance of livestock holders on crop residues, however caused for the shortage of livestock feed. During the discussions, the participants reported that the damage of crop fields by water hyacinth was one of the factors that accounted for the shortage of feed supply. Estimation made based on FAO [37] conversion factor indicated that a total of 159.8 tons of dry matter (DM) was lost due to the damage of rice fields. This could feed the existing cattle (1259 TLU) of the interviewed farmers for 25 days without supplementary feed. Interviewed farmers, who faced livestock feed shortage due to the damage of crop fields by the weed, have filled the feed gap by purchasing of crop residues (79.9%), hay (70.7%) and agro-industrial byproducts (25.6%). The descriptive statistics indicated that USD 73.32 /livestock holder spent to purchase hay, crop residues and agro-industrial byproducts.

The negative effect of the weed infestation on the communal grazing lands of the lakeshore and associated wetlands was also reported by *woreda* environmental protection offices of three selected woreda administrations. According to the reports of *woreda* environmental protection offices, the total area of communal grazing lands infested by the weed ranged from 37.4 sq.km (in 2016/17) to 203.9 sq.km (2014/16). The estimation made based on the conversion factors [37] revealed that 203.9 sq.km could supply about 40,780 tons of dry matter (DM). This was equivalent to 31.6% of the annual livestock feed requirement of the livestock holders of 21 infested KAs. In other words, the grazing pastures of lakeshore and wetlands damaged by the weed could support the existing stocks of 21 KA for four months without supplementary feed. Based on the local price of hay, the total price of the dry matter that could be harvested from the infested lakeshore and wetlands estimated to be USD 2, 718, 666.66.

**Table 4. Percentage distribution of households by the sources of water for various purposes.**

| Sources | Drinking water | Cooking | Livestock watering | Bathing | Washing cloths | Swimming |
|---|---|---|---|---|---|---|
| Protected dug well | 56 | 56 | 0.0 | 0.0 | 0.0 | 0.0 |
| Unprotected dug well | 580 | 58.0 | 36.0 | 56.0 | 58.0 | 0.0 |
| Unprotected spring | 12 | 12 | 0.0 | 0.0 | 0.0 | 0.0 |
| River/stream | 8 | 8 | 79.0 | 47 | 47 | 32 |
| Swamps | 0.0 | 0.0 | 710 | 32.0 | 320 | 15.0 |
| Lake | 0.0 | 0.0 | 40.0 | 39.0 | 38.0 | 560 |

## Impact of water hyacinth on water supply

The sources of water for drinking and cooking in the study areas were dug wells with hand pump and unprotected dug wells. The survey data indicated that 56.3% and 58.3% of households used water from the protected and unprotected dug well, respectively as their main source of drinking water and cooking. Lake Tana and its associated wetlands were directly or indirectly the sources of drinking water. Based on survey results, unprotected dug wells that were used as the source of drinking water and cooking quarried in the lakeshore and associated wetlands. As presented in summary Table 4, Lake Tana, river mouths and swamps were also the major sources of water for livestock watering, washing clothes, baths and washing.

Water hyacinth infestation in the shores of Lake Tana has affected water supply for various purposes. The participants of FGDs observed that the invasion of the lakeshore by water hyacinth has affected the water supply of drinking water, livestock watering and other uses by creating barrier to dig the shallow wells in the lakeshores and swampy lands, blocking wetlands and lakeshore water's access and deteriorating the quality of water by changing its odor.

The survey results also revealed the seriousness of the weed's effect on the water supply of the local people. More than 60% of households reported that the weed blocked digging of boreholes and fetching of water for drinking and cooking from the lake and river mouths. Livestock watering was being difficult for nearly 90% herders due to the blockage of the weed and poor odor of water. Almost all households reported that they could not swim in the lake and river mouths especially during the peak infestation seasons of water hyacinth. The dense mats of the weed impeded 70% of households to wash clothes and take showers in the lakeshore and river mouths. This is in agreement with the findings of [14] and [38] but contradictory with the results reported by [39] and [40]. [14] reported that water hyacinth had dropped the water supply in Kismu, Kenya by blocking water points and deteriorated water quality by making it muddy and changing its color. On the contrary, [39] and [40] reported that the weed contributed to the improvement of water quality due to its strong capacity to absorb nutrients and pollutants from eutrophic/polluted waters.

## Impact on human and livestock health

Since water hyacinth officially recognized as an ecologically dangerous invasive weed in the Lake shores, the local people have been participated in the weeding campaign (hand weeding). Based on the survey results, at least one family member of 92.8% of the interviewed households have been participated in manual removal of the weed. The average number of family members engaged in the weeding campaign was 1.43, ranged from 1 to 4. Their engagement in the manual removal campaign made the local people vulnerable to diseases and parasites. The participants of FGD in Achera KA explained the difficulty of manual removal of the dense mats of the weed and its effect on health of the people as:

*The residents of villages near and far to the lake participate in the manual removal of the emboch every year. We immerse deep into a water at least 6 hours a day to remove the mats. We do not dress any protective wears. The depth of water ranges from our knees to shoulders. Sometimes, the depth may be beyond our height. In this circumstance, the likelihood of individuals to be sink is high. Three persons lost their lives by sinking and then trapping by the dense mats of emboch during weeding. Removing of the dense mats takes place by pushing it from the deep water to the shore of the lake. Pushing the dense mats of emboch is tiresome. At least five people need to push a pile of mats. The water covered by the dense mats of emboch is extremely cold and smells bad. Waiting in a cold and bad smell water for six or more hours per day is troublesome and cause for skin disease. Everybody, who has been involved in manual removal, suffered with skin rush.*

The survey data corroborated the suffering of the people with skin rush and other diseases due to staying long hours per day in a cool and bad smell water. Almost all of respondents reported they were suffering with skin allergic (itching) while engaging in the manual removal of the weed. A similar result was reported by [33] in Southern Benin. De Groote et al (2003) reported that close of 1/3rd of the respondents was affected by the itching. An increasing the prevalence rate of malaria and bilharziosis that were reported in previous studies [28,33] were not observed by the respondents and participants of FGDs after the weed infested Lake Tana. However, the research done by [38] in the same study area revealed that the physical abstraction of water hyacinth provided a very good habitat for the proliferation of mosquito larvae. This is contradictory with the results reported in Nyanza Gulf of Lake Victoria by [41]. [41] indicated that the abundance of malaria causing mosquitoes associated with water hyacinth infestation in Nyanza Gulf accounted only 0.4% of the total number of mosquitoes. Unlike Lake Victoria Region [3], the mats of weed did not serve as a host for snakes and crocodiles.

Like human health, the health of the cattle has also affected because of feeding the weed. Based on survey results, the cattle of all of herders were affected by gut bloating and continuous diarrhea due to feeding water hyacinth. Concerning the prevalence rate of gut bloating and continuous diarrhea, about 1/3rd of cattle owners rated it as sever, 56.1% rated as moderate and the remaining 10.5% rated as low. The reason why the weed had resulted for the bloating of cattle gut and continuous diarrhea was not identified by FGDs and the interviewed farmers. [42] however cited consumption of the stalk tissues of water hyacinth which contain intercellular spaces filled with air as the principal factor for the bloating of ruminant guts and continuous diarrhea. According to [41], water hyacinth can be used for a healthy livestock feed if the weed chopped to eliminate intercellular spaces filled with air. In the study area, the cattle used the weed only by grazing. Livestock raisers have never been chopped the weed and fed the livestock.

An increasing the intensity of breeding of leech and other internal parasites was also observed after the weed infested the lake and associated marsh lands. The cattle of almost all owners were affected by leech and other internal parasites. The diseases and parasites that were caused by feeding water hyacinth has contributed to the loss of body weight and death of livestock. According to the survey results, the market sale of live animal has decreased because of the loss of the body weight at least by 1/3rd compared with the market sale of live animals in areas where the plant has not been infested. More than 1/10th of cattle owners reported the death of 1 to 3 heads by gut bloating and continuous diarrhea. About two-third of herders visited veterinary clinics for treatment of gut bloating and diarrhea. On average, they have spent 4.43 USD per head/year for veterinary services.

## The impact of water hyacinth on tourism and navigation

The Lake Tana area is one of important tourist sites in Ethiopia. The main natural attractions of tourist in the lake area are the forested peninsulas and islands, the birds and various historic heritages. The data taken from annual reports of Lake Tana Public Water Transport Enterprise revealed that on average 25,000 domestic and 6,000 foreign tourists have visited the natural landscapes and historic heritages of the lake from 2007 to 2017. Moreover, over 85,000 of local passengers and 7000 tons of goods per year were transported by the public water transport enterprise for ten years (2007 to 2017). The annual average income generated from transportation of goods, local passengers, domestic and foreign tourists was 0.35 million USD ranging from 0.13 to 0.52 million USD. In addition to Public Water Transport Enterprise, about 100 small private motor boat enterprises have provided touring services and transportation of goods. As estimated by the private boat enterprise cooperatives, the economic value of private enterprise navigation was 0.053 million USD per year.

The studies conducted in Nigeria and Blue River of Sudan [12,13] indicated that the dense mats of water hyacinth had affected water transportation by reducing the boat speed, increasing the working duration of engine per unit distance travelled and increasing the running and maintenance costs of the transportation. The key informants of both public and private water transport enterprises were asked whether or not the weed has brought similar problems on water transportation of Lake Tana. Based on KIIs findings, the weed's influence on navigation was minimum. According to them, most of the ports are found in the western part of Lake Tana. The lakeshore in the western part and most of the islands have not been infested by water hyacinth. The weed has infested only two ports such as Tana Cherkos and few parts of Gorgora. The transportation of tourists and goods has blocked by the weed particularly in Tana Chirkos. The large boats owned by the public enterprise could not access to the shore of this island. The key informants thought that the probability of water transport interrupted by water hyacinth would be high in the near future due to its rapid reproduction and the floating nature of its dense mats. According to them, the interruption of touring services and navigation by the weed would cost hundred thousands of USD that generated from tourism, and transportation of local passengers and goods and large number of people, who are employing in tourism sector, will lose their jobs The cost of water hyacinth infestation on tourism sector and water transportation was reported in previous studies [30,43]. As estimated by [43], water hyacinth infestation had cost about USD 2375 per day that was generated through tourism in Victoria Falls by drastically degrading the scenic views of the river and hindering the navigation. [30] also reported that the monthly income that was generated from water transport had dropped by 75% after the infestation of the water hyacinth in Wouri River Basin, Cameroon.

## Controls of water hyacinth

Different efforts have been done to control water hyacinth infestation since it has been officially recognized as an ecologically dangerous invasive weed in Lake Tana shores. Physical controlling mechanism (manual removal) has been the most important controlling mechanisms. The people, whose ages are 18 and above years, were obligated to take part in the campaign. Farmers were organized in the developmental group for the accomplishment of the weeding. Annually, the weeding campaign takes four to six months depends on the infestation level of the weed.

Based on the reports of EPFRWA, a large number of people have been participated in the hand weeding campaign since 2013/14. Fig 2 presents the number of person days participated in the manual removing and estimated cost from 2012/13 to 2016/17. On average, 157,000 persons days had involved in the hand weeding of water hyacinth in the consecutive five years. In general, close to 800,000 persons days had engaged to control water hyacinth infestation. As

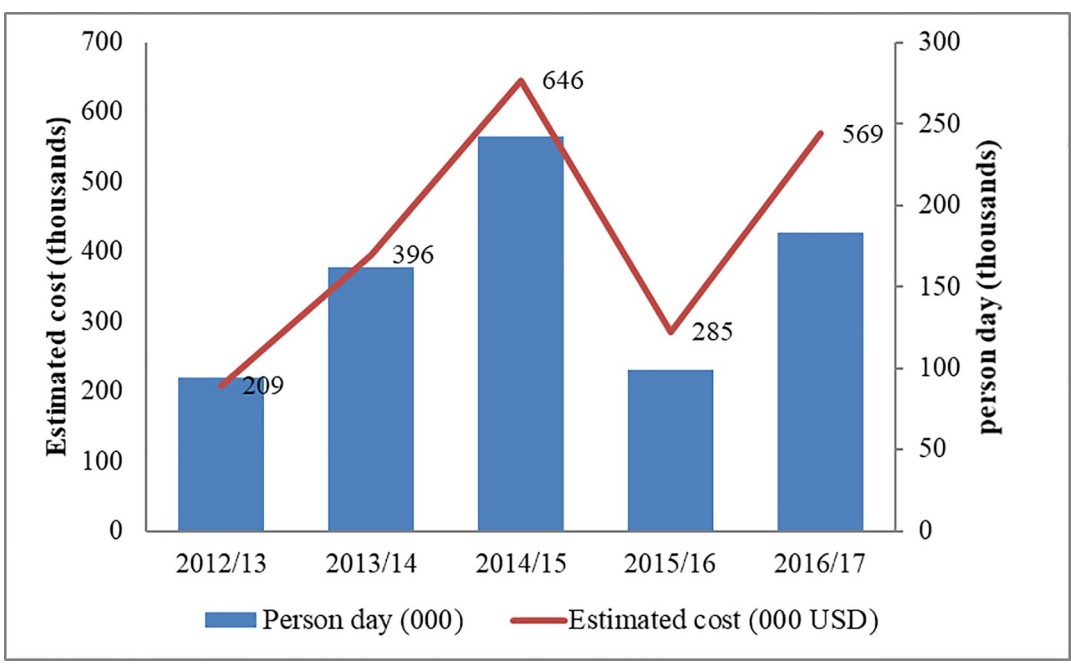

**Fig 2. The number of people participated in the weeding campaign of WH and estimated cost from 2012/13 to 2016/17.**

estimation made based on the local wage labor/day/person, the total cost of labor dedicated to manual removal from 2012/13 to 2016/17 was more than 2.1 million USD.

The manual removal campaign of water hyacinth has affected farming activities. Based on the survey results, about 87.3% of respondents reported that the weeding campaign has diverted their labor from farm activities to manual removal of water hyacinth. During discussions, the participants also stated that farmers did not able to practice weeding and harvesting of crops timely due to the involvement of all active labor force of family members in the weeding of water hyacinth for four days a week.

Similarly, off-and non-farm activities were affected by the weeding campaign. The survey data revealed that nearly 1/4th of respondents was involved in daily labor, petty trading and other non-farm activities. The data taken from the reports of *woreda* offices of micro and small-scale enterprise development (WoMSED) substantiated the contribution of off-and non-farm activities in employing a substantial number of people. According to WoMSED, about 1/3rd of households of 21 infested KAs were employed in off-and non-farm activities. These households were mostly young and they were either landless or own small size of farmland. Their participation in the weeding campaign has costed their time of earning income. Discussions made with focus groups reported that spending four days a week in water hyacinth controlling campaign has seriously affected the people whose life heavily relies on daily labor and other non-farm activities by compelling them to stop earning money that needs to fulfill the minimum requirement of daily life.

The weakening of social relationship of the community was another effect of the manual removal campaign of water hyacinth. The interviews made with chief administrators of KAs revealed that the bylaws have enacted to govern the participation of the people in the hand weeding campaign. Based on bylaws, an individual would be punished 1.85 USD per single day absence. The amount of money paid for absence of three or more days from the campaign was 3.73 USD per day. The findings of FGDs revealed that the enforcement of bylaws has created disputes between bylaw enforcers and the persons who were forced to partake in manual

removing. According to FGDs, the dispute ranged from insulting to beating bylaw enforcers. Sometimes, the dispute that is created between individuals are extended to group disputes. Such disputes are caused for the damage of social relationship of the disputants. Many numbers of people have withdrawn from social associations such as *iddir*, *mahber*, *senbete*, *debo* and *wonfel* due to the disputes created during the weeding campaign. As reported by FGDs, these social associations are crucial for mutual support, money lending, grain borrowing, agricultural activities and housing construction, and they had been maintained the social interactions among the community members for many decades. The substantial role of social associations to the strengthening of the rural community was also observed by [44] and [45].

Harvester machines have been used as well to control water hyacinth. Based on EFWPDA reports, four harvesting machines have procured by the support of private companies, charity organization and public universities. Close to 900,000 USD was spent for the purchasing and transportation of these harvesting machines. However, the evaluation made by the team of experts reported that all of harvesting machines were not operational. Technical problems of harvesters, lack of technical skills of operators, poor operational management and lack of spare parts were accountable for the non-functioning of harvester machines. The ineffectiveness of harvester machines in controlling water hyacinth was also reported in Lake Victoria [46,47].

Biological control of water hyacinth, which reported to be the most economical and sustainable method of control [48], has not yet been applied in Lake Tana. However, the use of bioagents particularly weevils have currently received attention. Researchers have conducted field trials to evaluate whether or not weevils are effective to eliminate the weed. Based on key informants, rearing of Neochetina weevils (*Neochetina eichhorniae* and *Neochetina bruchi*) has taken place for the use of bioagents against water hyacinth. For mass rearing of Neochetina weevils, lath houses have built. About USD 243,636.36 has allocated for the building of lath houses, rearing of mass weevils and conducting experiments.

Generally, more than 3.2 million USD have been invested and close to 800,000 labors have been involved in water hyacinth controlling efforts. The role of the controlling mechanisms undertaken so far however is not worth mentioning in eradicating water hyacinth. The findings of FGDs and KIIs cited three important reasons for the failure of manual removal campaign in controlling water hyacinth infestation. The poor coordination of weeding campaign is one of the most important factors for insignificant role of the campaign either to control the expansion or eradicate the weed. During the discussions, the participants reported that the KAs in which their lake shore area infested by water hyacinth had different weeding time schedule. Some KAs started the campaign early and others started lately. Furthermore, some KAs cleared the total area of land covered by the weed while others could not. In such situation, dense mats of the weed have been transported by the lake wave from lakeshore where the weed is not totally removed to the KAs that have been started the campaign early and removed the weed totally. This result is consistent with the findings of [8] which underlined that transportation of the mats by wind waves from one part of the lakeshore of Lake Victoria to other sides had made the weed controlling efforts very difficult.

Dumping of the harvested dense mats of the weed in the lakeshore is another important factor for less contribution of the weeding campaign in controlling the weed's fast expansion. During the discussions with FGs and key informants, it was reported that harvested mats of the weed piled in the lakeshore. Efforts have never been made either to avoid the piled mats from the lakeshore or to burn piles after weeding by the local administration or other concerned bodies. Dumping of the harvested dense mats of the weed thus contributed for re-colonization or further expansion of WH in lakeshore and wetlands. The regeneration of the weed was also reported by [49]. [49] reported that WH had rapidly regenerated in Lake Victoria after manual removal through hand pulling or using pitch forks.

Forced participation of the local people has also accounted for the failure of manual removal of water hyacinth in controlling its infestation. During the discussions, the participants reported that the local people have participated in manual removal campaign not for the sake of controlling water hyacinth infestation but to fulfil their obligation demanded by the government. It was also indicated in the survey data that the majority of the farmers considered the weeding campaign as the obligatory development work so that they have not been concerned about the quality of the work. The role of forced participation for the failure of natural resources management programs in Ethiopia was also reported by various authors [50–52]. These authors reported that the natural resources conservation works that were accomplished via forced community participation had failed because priority had given to quantity rather than quality or area coverage rather than impacts.

Considering eradication through physical, mechanical and biological as the only option may also contribute for the failure of weed's management in the study area. Efforts have not yet made to transform water hyacinth into economic, social and environmental benefits. During interview, key informants reported that only few NGOs particularly Nature and Biodiversity Union (NABU) organized the vulnerable groups and trained them to convert the weed into various benefits. [53] claimed that transforming water hyacinth into various benefits plays a role for sustainable management of the weed.

## Conclusions

This study assesses socioeconomic effects of water hyacinth since it has been officially recognized as an ecologically dangerous invasive weed in Lake Tana shores. The results revealed that the expansion of water hyacinth has affected the important resources of the lake that are basic to the livelihoods of the local people. The weed mats impacted crop production by damaging crop fields and making the farming activities difficult. The influence of water hyacinth on fishing was also serious. The catchability of fish has seriously decreased and the cost of fishing has increased by shortening the lifespan of fishing equipment since the weed infested the lakeshore. Besides, the weed has affected the use of lake water for various purposes by making quarrying of wells in lakeshore and swampy lands difficult, blocking water access and changing its odor. The weed has also affected livestock feed supply due to its damaging effects of various palatable grass species. Being the sole source of natural pasture by destructing the palatable grasses, the weed has increased the prevalence rate of livestock diseases and parasites. The expansion of the weed has also a serious implication on the national economy by disrupting hydropower generation and tourism. The range of socio-economic consequence of the weed generally implied the real impacts on the lives of local communities and national economic development. Since the start of the weed controlling efforts, close to 800 thousand of human labor devoted and more than 3.2 million USD has financed for the eradication of water hyacinth. The participation of local people in the manual removal campaign of the weed has exposed them to different diseases specifically skin allergy (itching). The devotion of huge number of labors and investment of significant amount of money have not yet played a role in controlling weed expansion. Poor coordination of controlling efforts, technical problems, dumping of harvested weed in the lakeshore and lack of genuine participation were identified as the principal factors for the failure of the controlling efforts of water hyacinth infestation. Furthermore, considering eradication mechanisms (manual removal, mechanical and biological methods) as the only option contributes for the failure of weed's management. The government has neglected the contribution of an extensive catchment-scale land use management and using the weed for various benefits for WH management in the Lake Tana shore. A coordination of various stakeholders is needed to make eradicating methods more effective. An

extensive catchment-scale land use management and the use of the weed into various benefits should also considered as the management option.

Even though this study provides information to the policy makers, executers and other stakeholders, it has some limitations that future studies need to address. The study is focused only on the overall socioeconomic effects of the weed. Detail investigations has not made on the impact of the weed in each provisioning ecosystem services provided by the lake ecosystem. Thus, we suggest further detail studies on the effect of the weed on each provisioning ecosystem service.

## Supporting information

**S1 Questionnaire. Household survey questionnaire for assessment of socioeconomic assessment of water hyacinth in Lake Tana shore.**
(PDF)

## Acknowledgments

We would like to acknowledge head of agricultural and environmental protection offices for providing information and facilitating administrative linkage with the sampled *kebele* administrations and development agents. Our thanks also go to the kebele administrative leaders for identifying the participants of FGD and supporting household survey. We owe distinguished appreciation to all farm households that have willingly devoted their time to share the challenges that they have faced by the infestation of water hyacinth.

## Author Contributions

**Conceptualization:** Belachew Getnet Enyew.

**Data curation:** Belachew Getnet Enyew, Workiyie Worie Assefa, Ayenew Gezie.

**Formal analysis:** Belachew Getnet Enyew.

**Investigation:** Belachew Getnet Enyew, Ayenew Gezie.

**Methodology:** Belachew Getnet Enyew, Workiyie Worie Assefa.

**Project administration:** Workiyie Worie Assefa.

**Validation:** Belachew Getnet Enyew.

**Writing – original draft:** Belachew Getnet Enyew.

**Writing – review & editing:** Belachew Getnet Enyew, Workiyie Worie Assefa, Ayenew Gezie.

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
