## [Decision Letter · Decision Letter 0]

28 May 2020

PONE-D-20-11958

Socioeconomic Effects of Water Hyacinth (Echornia Crassipes) in Lake Tana, North Western Ethiopia

PLOS ONE

Dear Dr. Eneyew,

Thank you for submitting your manuscript to PLOS ONE. After careful consideration, we feel that it has merit but does not fully meet PLOS ONE’s publication criteria as it currently stands. Therefore, we invite you to submit a revised version of the manuscript that addresses the points raised during the review process.

The manuscript by Eneyew et al. reports the socioeconomic impact of water hyacinth in Lake Tana, Ethiopia. I understand this invasive species is a huge problem in many parts of the world. A lot of research is underway on control efforts but it is important to highlight and quantify the real impact of such species on society and human welfare. Therefore, I found this manuscript timely and interesting. I was lucky enough to get it reviewed by two outstanding experts in the field who I think did a great job in critically assessing the quality of the manuscript. Based on the overall recommendation and detailed assessment provided by both reviewers I suggest a major revision before we can re-assess the manuscript. It is important to note that comments of both reviewers are coherent and mostly constructive which should be a great motivation and opportunity for the authors to improve their manuscript. Authors can see the detailed comments below. I must stress that I found all the critique relevant, therefore, authors must consider each and every comment/suggestion and provide detailed feedback. I also advise that all the revisions/corrections in the revised version be highlighted adequately and details should be also provided in a separate 'Response to Reviewers' document. Based on reviewers comments and my personal assessment of the manuscript, I would like to highlight the following broad points which authors should keep in mind while preparing the revision:

1. Comprehensively address/correct the issues raised around the methodology and survey techniques

2. Frame your hypotheses clearly as highlighted by Reviewer 2. It is important to discuss the objectives clearly and define that this study is not purely done in a social science context rather with major focus on invasion biology aspects.

3. Only explain the results based on the findings of the current survey and no assumptions or extrapolations should be made in results section.

4. Avoid using direct quotations from the survey responders as I its not a common practice in most management oriented publications and applied journals. Authors should report the quantitative results only and any inference or conclusions from interviews can be discussed on discussion section but briefly in their own words.

5. Both reviewers mentioned that authors have not made all the associated data available. I suggest to add relevant data as supplementary material or if that's not applicable and authors think all the data is within the manuscript then add a statement describing that.

6. Discussion is the weak link in the manuscript - it must highlight the logical reasoning for the current findings while providing a reasonable link with existing literature. Repeating results in discussion section and just referring to previous studies is not appropriate.

7. Revise conclusions in line with results and provide a concise message for scientific community as well as other stakeholders.

8. Reduce the overall length of the article by avoiding repetition.

9. There are several grammar/language errors throughout the manuscript and it is difficult for us to point out all of them. I suggest authors organize the revised version to proofread by a native English speaker.

We look forward to receiving your revised manuscript.

Kind regards,

Ali Ahsan Bajwa, Ph.D

Academic Editor

PLOS ONE

Journal Requirement:

3.  Please ensure that you refer to Figure 1 in your text as, if accepted, production will need this reference to link the reader to the figure

4. Please upload a copy of Figure 2, to which you refer in your text on page xx. If the figure is no longer to be included as part of the submission please remove all reference to it within the text.

5. We note you have included a table to which you do not refer in the text of your manuscript. Please ensure that you refer to Tables 1 and 3 in your text; if accepted, production will need this reference to link the reader to the Table.

Reviewers' comments:

Reviewer's Responses to Questions

**Comments to the Author**

1. Is the manuscript technically sound, and do the data support the conclusions?

Reviewer #1: Partly

Reviewer #2: Partly

2. Has the statistical analysis been performed appropriately and rigorously? 

Reviewer #1: No

Reviewer #2: I Don't Know

3. Have the authors made all data underlying the findings in their manuscript fully available?

Reviewer #1: No

Reviewer #2: No

4. Is the manuscript presented in an intelligible fashion and written in standard English?

Reviewer #1: Yes

Reviewer #2: Yes

5. Review Comments to the Author

Reviewer #1: General comments

The manuscript describe socio-economic impacts and management efforts of water hyacinth, one of the most notorious invasive weeds of inland freshwater wetlands, in Lake Tana – the largest lake in Ethiopia. The authors used a range of social survey tools to collect primary data. The results showed that impact of water hyacinth on local livelihood is significant, and the weed continue to spread/persist in spite of huge efforts made to remove it from the lake. It shows how damaging are invasive species to local livelihood and how difficult is it to manage once such species establish in vulnerable ecosystems. Therefore, the manuscript merit publication. But, the manuscript should be revised thoroughly before it can be considered for publication. I have mentioned specific comments below. General comments are:

1) The manuscript length is very long. Particularly the result section can be shortened. Authors presented several quotes of the focused group discussion in the result section. While this is a common practice in social science research, this is not common in natural science research. Some of these text can be avoided by highlighting the main message in a few sentences.

2) What is presented as Result and Discussion in the manuscript is simply the Result. Only at a few instances, authors mentioned that their results were similar to previous research. And, this is not discussion. I suggest to rewrite Results section and add a concise Discussion section.

Specific comment are:

Line 18: Mention the number of group discussions and key informants.

Line 19-20: Are the impacts arranged in decreasing order, i.e. one with the highest impact to the one with lowest impact? If yes, that's fine. If not, re-arrange accordingly.

Line 26-27: What is the barrier for the success of control?

Line 36: Start introduction with a broad overview of the problem of biological invasions in general and this problem in freshwater aquatic/wetland ecosystem in particular.

Line 44-45: If there is any information about dispersal pathways (how WH might have reached to the lake?), mention it here with reference.

Line 46-47: Official recognition of Eichhornia as ecological threat is different from the initiation of invasion by the species. Often problem is recognized by regulatory bodies long after the actual initiation of the problem. Rephrase this sentence to make it clear.

Line 50: Convert hectare into sq m or sq km. Do this throughout the manuscript.

Line 52: Unit of area missing.

Line 67-71: Also mention here that any effort to use the biomass of Eichhornia in the introduced range is a part of integrated management of Eichhornia. However the amount of biomass used is small and this practices have not helped much in the control of Eichhhornia. Rather, the practices have increased the chance of its introduction to areas. You may also mention that the socio-economic benefit of Eichhornia is insignificant compared to the socio-ecological damages that it has incurred.

Line 72: What is WH? Probably, water hyacinth but mention it when used first.

Line 87-90: Specifically mention objective, methodological approaches and implications of the results in the last paragraph of the introduction.

Line 93: Add elevation.

Line 95: Mention mean annual rainfall.

Line 112: This map is not meaningful. I suggest to include map of Ethiopia with location of the lake highlighted. Then include another topographic map showing details of the lake area including tributaries of the lake and Blue Nile river.

Line 124: Include reference for this formula.

Line 132-137: Mention specific source of information as references. For example, Tourism Development Office may have annual report. Need to explain the exact source of information. If authors have collected information by interviewing key authorities of these institutions, this should be mentioned as personal communication with name of the person, address and date of communication.

Line 143: Include questionnaire as Supplementary information.

Line 156: How may key informants were interviewed?

Line 182-185: In addition also mention how the impacts of WH in term of the amount of crop loss and monetary value were calculated? Also elaborate ANOVA. Which parameters were compared? Which statistical software was used? Were the data checked for normality and homoscedasticity before ANOVA?

Line 199: Though the title of this section is 'Results and Discussion', there is virtually no discussion. Just stating that some of the findings of the authors are similar the findings reported previously is not discussion. I suggest to present Results and Discussion separately so that a concise Discussion section can be included. Length of the Result section also need to be reduced.

Line 202-203: Give SD value for age of respondents and family size.

Line 207: Do not give two values in percentage. You can say '..... nearly 4/5th of the households (78.2%)'.

Line 212: What were these off and non-farm activities?

Line 238-239: How did authors categorized infestation as sever, moderate or scattered? Mention this in method section. I suggest to replace 'infestation rate' by 'infestation level'. The 'rate' is often used in temporal scale.

Line 251-252: I suggest to use SI system of unit throughout the manuscript.

Line 269: Replace 'plant' by 'weed'.

Line 281: Also present value in USD for the convenience of international readers.

Line 290: In the table, also mention Degree of Freedom along with F values.

Line 389-297: The quote is not essential. As the manuscript if focused to WH, the quoted text does not have anything about WH.

Line 447: Table description is not clear. What do the numbers in the table indicate?

Line 452-460: This quote is again not essential and can be avoided.

Line 560: Manual and mechanical removal of biomass of WH was mentioned as the major control activities in the Lake Tana. There was no mention about post-harvest processing of the biomass of WH. In many instances, collected biomass is dumped at the lake shore from where WH again re-colonize the wetlands. Therefore, give a brief explanation on how harvested biomass of WH was used.

Line 578: What is the monetary unit of cost?

Line 597-609: Length of this quote can be shortened to have by deleting background information.

Line 620-630: This can be avoided and information summarized in a few sentences.

Line 648: Give Latin names of the weevils under evaluation.

Line 655-683: The result section mainly summarized the results. While linking with results, the Conclusion section should also be able to present future outlook such as research areas, management interventions, policy changes etc.

Reviewer #2: The paper contains many interesting details about impacts of water hyacinth, but has not provided a sufficient description of methods to be considered a rigorous scientific study. The goal, to include a broad range of social and economic impacts, requires that the authors be judicious with details, which is both the problem and the key to improving the manuscript. For example, the manuscript includes a lot of detail on farming practices that is not directly relevant to the issue of determining harm from water hyacinth. The authors have included details that enrich the social and anthropological understanding of the community but also seem to be framing this analysis as an economic study, in which such detail would typically be omitted. However, when judged from either an anthropology or economics perspective, the methods and results are incomplete and the some of the economic estimates require additional considerations to be made robust.

A scientific manuscript (in any discipline) should have sufficient detail to make it replicable, at least to a large degree. Here, the methods are not specific enough for anyone to attempt to replicate this approach nor to judge whether the methods were rigorous and objective. I provide more details about what is missing below.

The results section also lacks key details about how to interpret results. For example, it is not clear whether numeric results are being interpreted in terms of the surveyed population or the population of the geographic regions. Non-responses are not discussed. The data were collected by region, but differences or similarities among regions were not discussed. The report also incorporates a great deal of government data with no specific citations. The English used is quite good but contains numerous grammatical errors that I did not correct. These grammatical errors are understandable given the difficulty of some English verb conjugations and the authors have my sympathies.

Introduction

1. Details of the area of WH infestation belong in case study description, not introduction.

2. Citation 11 is a conference presentation. The statement about mosquito-borne diseases requires a peer-reviewed citation. Evidence is mixed for the human health effects as a general rule, although the relationship may well be true for the case study area. See e.g., Ofulla, A.V.O., Karanja, D., Omondi, R., Okurut, T., Matano, A., Jembe, T., Abila, R., Boera, P., Gichuki, J., 2010. Relative abundance of mosquitoes and snails associated with water hyacinth and hippo grass in the Nyanza gulf of Lake Victoria. Lakes & Reservoirs: Science, Policy and Management for Sustainable Use 15, 255–271. https://doi.org/10.1111/j.1440-1770.2010.00434.x

3. Provide scientific name for papyrus

Methods

Methods are incomplete in almost all cases because they do not provide sufficient specific detail.

1. To better orient the reader, prior to the sampling procedures heading explain basic facts about the survey. What type? Goals of the survey (use and nonuse values?). Was the survey the only instrument used? Also, explain the goals of the other data collection (costs of management?).

2. Define what period is considered "before" and "after" infestation, as used in later analyses. How were these periods defined?

3. Provide a references for clustered systematic sampling technique and equation. Explain e before mentioning the size of N to finish explaining the equation. Define what you mean by precision.

4. Given the large number of data types and sources, I suggest creating a table of data types and sources and how they were used. For example, data could be categorized by the type of benefit or cost they describe.

Household survey

1. How were the houses identified? Provide copy of the questionnaire.

2. Briefly describe the type of socio-demographic data and farm characteristics that you sought to collect from respondents.

3. Be specific about the survey pre-test. How many interviews were conducted? Did they cover multiple regions?

4. If those conducting the survey were drawn from the local region, say so.

5. Enumerator is usually only used for a population census. Here I would say surveyor.

6. Be specific about what you mean by "regularly monitored". Did you observe them? What % of interviews were monitored and confirmed?

Key informant interviews

Explain the goals of the key informant interviews. How many informants per region?

Focus groups

Explain goals of focus group discussions. Explain how discussions were structured.

Methods of data analysis

1. When you describe ANOVA - What were you comparing? When ANOVA results are presented, say clearly and explicitly what you were testing and over what population you were testing your hypotheses.

2. You say the qualitative data were “analyzed based on themes.” Explain how themes were identified and any systematic coding that may have been conducted.

3. How did farmers estimate the area that they could not farm? Did the interviewer help them make the estimate? Or, did they have records of prior year area?

4. Provide more details on how additional labor rates were estimated. Was this based on records or recollections? How many farms' data were used in making such estimates? How many years back were you asking people to estimate? This information is relevant to understanding the reliability of the data.

Results

State whether you are providing results solely for the surveyed set or whether you have estimated results for the population of the regions.

It would be typical to create a table of the most relevant demographic and socioeconomic characteristics providing mean, range, national average.

Details of crop production that are not relevant to the impacts of water hyacinth can be summarized in a table rather than discussed in text.

Most numbers should be rounded to integer values.

Define crops being “totally damaged” Was any amount harvested? If no, then say crop was lost. Otherwise, describe yield or % harvested. Or, describe whether area was never planted as this has different economic implications.

Lines 262-265 - Justify the use of the comparison harvest. Cite the source of the data. Is it an average yield? Could weather have been to blame rather than water hyacinth?

253-354 - If farmers did not plant, they saved money on seed and labor. Can you estimate the net cost?

Line 290 - include sample size (n) in table

327-333 - more information needed. How were these figures calculated? What was the trend in catch prior to infestation.

346 - How, exactly, does the weed shorten the boat service time? Does it damage the boat?

All data that were used from government reports or offices needs to have a citation.

411 – move the information about why water hyacinth does not make good fodder to this section. Otherwise, it isn’t clear why you need alternative fodder.

418-420 - Why did you set the price based on hay? Is it an equivalent or a substitute? Be specific about how to appropriately estimate this value, given how farmers adapted. I suggest that you use the information from line 436 to estimate damages since it includes how farmers adapted.

447 - What are the units in this table? If percentages, why don't columns sum to 100?

548 - explain that officials were worried about the future impact of water hyacinth early in the paragraph. Otherwise, it isn't clear why you are providing detailed information about tourism when you say effects are minimal.

611 -The information about mandatory participation should be mentioned during the discussion of survey results about participation.

Conclusions

The last two sentences are not clearly conclusions of the analysis. I suggest trying to provide an improved quantitative summary of the many impacts that would speak for itself.

6. PLOS authors have the option to publish the peer review history of their article (what does this mean?). If published, this will include your full peer review and any attached files.

Reviewer #1: Yes: Bharat Babu Shrestha, Central Department of Botany, Tribhuvan University, Kathmandu, Nepal

Reviewer #2: No

---

## [Author Response · Author response to Decision Letter 0]

18 Jun 2020

A rebuttal letter

The comments given by both academic editor and reviewers are very constructive and we believe that they helped us for the improvement of the manuscript. We have tried to give responses for each point that were raised by academic editor and reviewers. The responses are reported as follows.

Academic editor

1. Comprehensively address/correct the issues raised around the methodology and survey techniques

Response: correctios have done based on the comments given by reviewers

2. Frame your hypotheses clearly as highlighted by Reviewer 2. It is important to discuss the objectives clearly and define that this study is not purely done in a social science context rather with major focus on invasion biology aspects.

3. Only explain the results based on the findings of the current survey and no assumptions or extrapolations should be made in results section.

Response: conclusion is revised based the given comment

4. Avoid using direct quotations from the survey responders as I is not a common practice in most management-oriented publications and applied journals. Authors should report the quantitative results only and any inference or conclusions from interviews can be discussed on discussion section but briefly in their own words.

Response: done based on the common

5. Both reviewers mentioned that authors have not made all the associated data available. I suggest to add relevant data as supplementary material or if that's not applicable and authors think all the data is within the manuscript then add a statement describing that.

Response: Household survey questionnaire is added as a supplementary material in revised submission. Other associated data will be submitted if necessary and requested. We believe the results reported in the manuscript are educate to provide information

6. Discussion is the weak link in the manuscript - it must highlight the logical reasoning for the current findings while providing a reasonable link with existing literature. Repeating results in discussion section and just referring to previous studies is not appropriate.

Response: We accepted the comments and have tried to incorporate the discussions

7. Revise conclusions in line with results and provide a concise message for scientific community as well as other stakeholders.

Response: the conclusion is modified based on the comments

8. Reduce the overall length of the article by avoiding repetition.

Response: The issues/quotations that are not necessary to the paper are deleted. However, the deletion of some unnecessary paragraphs or sentences cannot able to shorten the length of the article. Detail explanation done for the questions raised by reviewers rather increase the number of pages and words

9. There are several grammar/language errors throughout the manuscript and it is difficult for us to point out all of them. I suggest authors organize the revised version to proofread by a native English speaker.

Response: W e have tried to correct grammatical/language errors. However, proofreading by a native English speaker is not possible within the given time

Reviewer 1

General Comments

1) The manuscript length is very long. Particularly the result section can be shortened. 

Response: we have tried to shortened the result section by deleting the questions which are not much related to the paper based on the comment. However, he requested detail explanation for some issues, we add paragraphs and sentences. Besides, discussions incorporated based on the request. Thus, the length of the manuscript remained the same or increased

2) What is presented as Result and Discussion in the manuscript is simply the Result. Only at a few instances, authors mentioned that their results were similar to previous research. And, this is not discussion. I suggest to rewrite Results section and add a concise Discussion section

Response: We accepted that discussions have not made adequately in the submitted manuscript and the revised manuscript incorporate the discussions. However, results and discussions are not separated. 

Specific comment 

1. Line 18: Mention the number of group discussions and key informants.

Response: mentioned

2. Line 19-20: Are the impacts arranged in decreasing order, i.e. one with the highest impact to the one with lowest impact? If yes, that's fine. If not, re-arrange accordingly.

Response: Yes

3. Line 26-27: What is the barrier for the success of control?

Response: incorporated in the revised manuscript

4. Line 36: Start introduction with a broad overview of the problem of biological invasions in general and this problem in freshwater aquatic/wetland ecosystem in particular.

Response: Done based on comment

5. Line 44-45: If there is any information about dispersal pathways (how WH might have reached to the lake?), mention it here with reference.

Response: there is no information about how WH reached to the lake 

6. Line 46-47: Official recognition of Eichhornia as ecological threat is different from the initiation of invasion by the species. Often problem is recognized by regulatory bodies long after the actual initiation of the problem. Rephrase this sentence to make it clear.

Response: Rephrased 

7. Line 50: Convert hectare into sq m or sq km. Do this throughout the manuscript.

Response: Done

8. Line 52: Unit of area missing.

Response: the area is incorporated in the revised manuscript

9. Line 67-71: Also mention here that any effort to use the biomass of Eichhornia in the introduced range is a part of integrated management of Eichhornia. However, the amount of biomass used is small and this practice have not helped much in the control of Eichhhornia. Rather, the practices have increased the chance of its introduction to areas. You may also mention that the socio-economic benefit of Eichhornia is insignificant compared to the socio-ecological damages that it has incurred.

Response: correction has done based on the comments

10. Line 72: What is WH? Probably, water hyacinth but mention it when used first.

Response: mentioned

11. Line 87-90: Specifically mention objective, methodological approaches and implications of the results in the last paragraph of the introduction.

Reponses: 

12. Line 93: Add elevation.

Response: Done

13. Line 95: Mention mean annual rainfall.

Response: Done

14. Line 112: This map is not meaningful. I suggest to include map of Ethiopia with location of the lake highlighted. Then include another topographic map showing details of the lake area including tributaries of the lake and Blue Nile river.

Response: The map has changed and we develop a map based on request

15. Line 124: Include reference for this formula.

Response: the reference is incorporated in revised version

16. Line 132-137: Mention specific source of information as references. For example, Tourism Development Office may have annual report. Need to explain the exact source of information. If authors have collected information by interviewing key authorities of these institutions, this should be mentioned as personal communication with name of the person, address and date of communication.

Response: done based on comment

17. Line 143: Include questionnaire as Supplementary information.

Response: the questionnaire is attached for supplementary information

18. Line 156: How may key informants were interviewed?

Response: the number is incorporated in the revised version

19. Line 182-185: In addition, also mention how the impacts of WH in term of the amount of crop loss and monetary value were calculated? Also elaborate ANOVA. Which parameters were compared? Which statistical software was used? Were the data checked for normality and homoscedasticity before ANOVA?

Response: Done

20. Line 199: Though the title of this section is 'Results and Discussion', there is virtually no discussion. Just stating that some of the findings of the authors are similar the findings reported previously is not discussion. I suggest to present Results and Discussion separately so that a concise Discussion section can be included. Length of the Result section also need to be reduced.

Response: the discussions are incorporated together with the results

21. Line 202-203: Give SD value for age of respondents and family size.

Response: Done

22. Line 207: Do not give two values in percentage. You can say '..... nearly 4/5th of the households (78.2%)'.

Response: correction is made based on the comment

23. Line 212: What were these off and non-farm activities?

Response: mentioned

24. Line 238-239: How did authors categorized infestation as sever, moderate or scattered? Mention this in method section. I suggest to replace 'infestation rate' by 'infestation level'. The 'rate' is often used in temporal scale.

Response: correction is made based on the given comment

25. Line 251-252: I suggest to use SI system of unit throughout the manuscript.

Response: Done 

26. Line 269: Replace 'plant' by 'weed'.

Response: replaced

27. Line 281: Also present value in USD for the convenience of international readers.

Response: done

28. Line 290: In the table, also mention Degree of Freedom along with F values.

Response: incorporated

29. Line 389-297: The quote is not essential. As the manuscript if focused to WH, the quoted text does not have anything about WH.

Response: Deleted

30. Line 447: Table description is not clear. What do the numbers in the table indicate?

Response: The title of the table was vague and it is corrected. 

31. Line 452-460: This quote is again not essential and can be avoided.

Response: avoided

32. Line 560: Manual and mechanical removal of biomass of WH was mentioned as the major control activities in the Lake Tana. There was no mention about post-harvest processing of the biomass of WH. In many instances, collected biomass is dumped at the lake shore from where WH again re-colonize the wetlands. Therefore, give a brief explanation on how harvested biomass of WH was used.

Response: explanation has done based on the comment

33. Line 578: What is the monetary unit of cost?

Response: USD (incorporated based on the request)

34. Line 597-609: Length of this quote can be shortened to have by deleting background information.

Response: done based on the comment

35. Line 620-630: This can be avoided and information summarized in a few sentences.

Response: the quote is avoided and the information is summarized

36. Line 648: Give Latin names of the weevils under evaluation.

Response: Done

37. Line 655-683: The result section mainly summarized the results. While linking with results, the Conclusion section should also be able to present future outlook such as research areas, management interventions, policy changes etc.

Response: Modification is done based on the given comment

Reviewer 2

1. Details of the area of WH infestation belong in case study description, not introduction

Response: The figures about infested area are already reported by others not so that we prefer to report in the introduction part to highlight the problem

2. Citation 11 is a conference presentation. The statement about mosquito-borne diseases requires a peer-reviewed citation. Evidence is mixed for the human health effects as a general rule, although the relationship may well be true for the case study area. See e.g., Ofulla, A.V.O., Karanja, D., Omondi, R., Okurut, T., Matano, A., Jembe, T., Abila, R., Boera, P., Gichuki, J., 2010. Relative abundance of mosquitoes and snails associated with water hyacinth and hippo grass in the Nyanza gulf of Lake Victoria. Lakes & Reservoirs: Science, Policy and Management for Sustainable Use 15, 255–271. https://doi.org/10.1111/j.1440-1770.2010.00434.x

Response: it is reviewed

3. Provide scientific name for papyrus

Response: done

Methods

Methods are incomplete in almost all cases because they do not provide sufficient specific detail.

4. To better orient the reader, prior to the sampling procedures heading explain basic facts about the survey. What type? Goals of the survey (use and nonuse values?). Was the survey the only instrument used? Also, explain the goals of the other data collection (costs of management?).

Response: Correction is made based on the comments

5. Define what period is considered "before" and "after" infestation, as used in later analyses. How were these periods defined?

Response: As already reported in the introduction part, the weed has observed in 2010 and controlling efforts have started in 2011. Before refers to the period before 2011. And “after’ infestation refers to after 2011

6. Provide a reference for clustered systematic sampling technique and equation. Explain e before mentioning the size of N to finish explaining the equation. Define what you mean by precision.

Response: Correction is made based on the comment

7. Given the large number of data types and sources, I suggest creating a table of data types and sources and how they were used. For example, data could be categorized by the type of benefit or cost they describe.

Response: We believe that creation of extra table will increase the length of manuscripts

Household survey

8. How were the houses identified? Provide copy of the questionnaire.

Response: how the households identified is reported in revised version

9. Briefly describe the type of socio-demographic data and farm characteristics that you sought to collect from respondents.

Response: reported in the revised version

10. Be specific about the survey pre-test. How many interviews were conducted? Did they cover multiple regions?

Response: done based on request

11. If those conducting the survey were drawn from the local region, say so.

Response: Yes

12. Enumerator is usually only used for a population census. Here I would say surveyor.

Response: Enumerator changed to data collector

13. Be specific about what you mean by "regularly monitored". Did you observe them? What % of interviews were monitored and confirmed?

Response: correction is done 

Key informant interviews

14. Explain the goals of the key informant interviews. How many informants per region?

Response: done

Focus groups

15. Explain goals of focus group discussions. Explain how discussions were structured.

Response: the goal is reported and the structure of FGDs incorporated

Methods of data analysis

16. When you describe ANOVA - What were you comparing? When ANOVA results are presented, say clearly and explicitly what you were testing and over what population you were testing your hypotheses.

Response: The variables that were compared are explained

17. You say the qualitative data were “analyzed based on themes.” Explain how themes were identified and any systematic coding that may have been conducted.

Response: Systematic coding has not conducted. Themes were identified simply from transcription

18. How did farmers estimate the area that they could not farm? Did the interviewer help them make the estimate? Or, did they have records of prior year area?

Response: Farmers know their farm area that is damaged by the weed. Estimation is thus done based on the data obtained from them

19. Provide more details on how additional labor rates were estimated. Was this based on records or recollections? How many farms' data were used in making such estimates? How many years back were you asking people to estimate? This information is relevant to understanding the reliability of the data.

Response: 

Results

20. State whether you are providing results solely for the surveyed set or whether you have estimated results for the population of the regions.

Response: both the surveyed and population

21. It would be typical to create a table of the most relevant demographic and socioeconomic characteristics providing mean, range, national average.

Response: incorporating the requested table will increase the length and we omitted it. 

22. Details of crop production that are not relevant to the impacts of water hyacinth can be summarized in a table rather than discussed in text.

Response: in our opinion, detail report is note done concerning crop production which is not relevant to the study. 

23. Most numbers should be rounded to integer values.

Response: 

24. Define crops being “totally damaged” Was any amount harvested? If no, then say crop was lost. Otherwise, describe yield or % harvested. Or, describe whether area was never planted as this has different economic implications.

Response: we did see the problem. It is clearly reported

25. Lines 262-265 - Justify the use of the comparison harvest. Cite the source of the data. Is it an average yield? Could weather have been to blame rather than water hyacinth?

 Response: the cropping seasons mention in the report were normal harvesting season. Weather did not influence crops production. We think that report does not confuse the readers. It is clear

26. 253-354 - If farmers did not plant, they saved money on seed and labor. Can you estimate the net cost?

Response: farmers grew rice but the rice field was damaged. So, the family labour was dedicated for farm activities. Seeds were also sown

27. Line 290 - include sample size (n) in table

Response: If the table about the sample size is incorporated, we believe that the length of manuscript will be increased. So, the verbal explanation is adequate

28. 327-333 - more information needed. How were these figures calculated? What was the trend in catch prior to infestation?

Response: correction is made based on request

29. 346 - How, exactly, does the weed shorten the boat service time? Does it damage the boat?

Response: Yes

30. All data that were used from government reports or offices needs to have a citation.

Response: done

31. 411 – move the information about why water hyacinth does not make good fodder to this section. Otherwise, it isn’t clear why you need alternative fodder.

Response: correction is done based on comment

32. 418-420 - Why did you set the price based on hay? Is it an equivalent or a substitute? Be specific about how to appropriately estimate this value, given how farmers adapted. I suggest that you use the information from line 436 to estimate damages since it includes how farmers adapted.

Response: the farmers have harvested hay from the communal pasture lands of the lakeshore and wetlands. That is why estimation was done based on the price of hay

33. 447 - What are the units in this table? If percentages, why don't columns sum to 100?

Response: correction is done

34. 548 - explain that officials were worried about the future impact of water hyacinth early in the paragraph. Otherwise, it isn't clear why you are providing detailed information about tourism when you say effects are minimal.

Response: explained

35. 611 -The information about mandatory participation should be mentioned during the discussion of survey results about participation.

Response: Explained

Conclusions

36. The last two sentences are not clearly conclusions of the analysis. I suggest trying to provide an improved quantitative summary of the many impacts that would speak

Response: conclusions have modified based on the given comments

---

## [Decision Letter · Decision Letter 1]

14 Jul 2020

PONE-D-20-11958R1

Socioeconomic Effects of Water Hyacinth (Echornia Crassipes) in Lake Tana, North Western Ethiopia

PLOS ONE

Dear Dr. Eneyew,

Thank you for submitting your manuscript to PLOS ONE. After careful consideration, we feel that it has merit but does not fully meet PLOS ONE’s publication criteria as it currently stands. Therefore, we invite you to submit a revised version of the manuscript that addresses the points raised during the review process.

ACADEMIC EDITOR: Comments below.

We look forward to receiving your revised manuscript.

Kind regards,

Ali Ahsan Bajwa, Ph.D

Academic Editor

PLOS ONE

Additional Editor Comments (if provided):

Authors have revised the manuscript substantially. The Reviewer 2 was not available to review the revision but I can see authors have addressed most of their his/her comments. The Reviewer 1 also noted significant improvement, however, there are still a few issues that require revision before we can assess the manuscript further. Therefore, I request authors to undertake a comprehensive revision paying full attention to further comments. I also suggest to further reduce the overall length of manuscript.

Reviewers' comments:

Reviewer's Responses to Questions

**Comments to the Author**

1. If the authors have adequately addressed your comments raised in a previous round of review and you feel that this manuscript is now acceptable for publication, you may indicate that here to bypass the “Comments to the Author” section, enter your conflict of interest statement in the “Confidential to Editor” section, and submit your "Accept" recommendation.

Reviewer #1: (No Response)

2. Is the manuscript technically sound, and do the data support the conclusions?

Reviewer #1: Yes

3. Has the statistical analysis been performed appropriately and rigorously? 

Reviewer #1: Yes

4. Have the authors made all data underlying the findings in their manuscript fully available?

Reviewer #1: Yes

5. Is the manuscript presented in an intelligible fashion and written in standard English?

Reviewer #1: Yes

6. Review Comments to the Author

Reviewer #1: General comments

The manuscript has been substantially improved but there are several minor issues to be fixed before the manuscript can be considered for publication. There are still several quotes of FGDs in the manuscript. Reviewer and editor suggested to delete them and include main message in running text. This will make the manuscript concise.

Some specific comments are below:

Line 46: It is good to use IAPS instead of IAS for the Invasive Alien Plant Species.

Line 107-109: Geographic range (latitude and longitude) given in the text and map (Figure 1) does not match. It appears that the text description is true for Lake Tana Basin Boundary, rather than Lake Tana. Please check and correct. Replace zero in superscript by symbol degree in latitude and longitude values.

Line 129: Scale bar in two maps seems misplaced/exchanged. Please check scale bar of each map and correct them.

Line 156: The word 'about' is probably incorrect. If 408 is the exact number of households surveyed, write this in word without 'about' here.

Line 278: Table 1- What is HHs in second column head? Spell out as there is no space limitation.

Line 286-287: It is good to follow SI unit system for yield value. Please present yield values in round figure, e.g. 3052 instead of 3052.35. Do this rounding for all similar values (e.g. in line 293).

Line 414: Spell TLU in the last column.

Line 478: Table 4 - present all values in round figure without decimal.

Line 604-605: Check figure number. You already have Fig 1 in study area section. Make title of both y axis clear. Include 'Thousands' inside parentheses.

Line 701-740: Length of conclusion can be shortened by deleting texts that repeat results.

7. PLOS authors have the option to publish the peer review history of their article (what does this mean?). If published, this will include your full peer review and any attached files.

Reviewer #1: **Yes: **Bharat Babu Shrestha

---

## [Author Response · Author response to Decision Letter 1]

29 Jul 2020

Reviewer 1

1. There are still several quotes of FGDs in the manuscript. Reviewer and editor suggested to delete them and include main message in running text. This will make the manuscript concise.

Response: Four out of five quotes of FGDs in the manuscript are deleted and included in the main message in running text. We believe that one of FGDs quotes is put as it is to provide a picture on the issue.

2. Line 46: It is good to use IAPS instead of IAS for the Invasive Alien Plant Species.

Response: IAS changed to IAPS

3. Line 107-109: Geographic range (latitude and longitude) given in the text and map (Figure 1) does not match. It appears that the text description is true for Lake Tana Basin Boundary, rather than Lake Tana. Please check and correct. Replace zero in superscript by symbol degree in latitude and longitude values. 

Response: The geographical locations given in the map are written in text . Zero is replaced by symbol

4. Line 129: Scale bar in two maps seems misplaced/exchanged. Please check scale bar of each map and correct them.

Response: We made correction

5. Line 156: The word 'about' is probably incorrect. If 408 is the exact number of households surveyed, write this in word without 'about' here.

Response: Correction is made based on comments

6. Line 278: Table 1- What is HHs in second column head? Spell out as there is no space limitation.

Response: Done based on comments

7. Line 286-287: It is good to follow SI unit system for yield value. Please present yield values in round figure, e.g. 3052 instead of 3052.35. Do this rounding for all similar values (e.g. in line 293).

Response: corrections have made based on comments

8. Line 414: Spell TLU in the last column.

Response: TLU is spelt use Tropical Livestock Unit

9. Line 478: Table 4 - present all values in round figure without decimal.

Response: done 

10. Line 604-605: Check figure number. You already have Fig 1 in study area section. Make title of both y axis clear. Include 'Thousands' inside parentheses.

Response: Corrected based on comments

11. Line 701-740: Length of conclusion can be shortened by deleting texts that repeat results.

Response: we tried to shortened

---

## [Editor Report · Decision Letter 2]

31 Jul 2020

Socioeconomic Effects of Water Hyacinth (Echornia Crassipes) in Lake Tana, North Western Ethiopia

PONE-D-20-11958R2

Dear Dr. Eneyew,

We’re pleased to inform you that your manuscript has been judged scientifically suitable for publication and will be formally accepted for publication once it meets all outstanding technical requirements.

Kind regards,

Ali Bajwa, Ph.D

Academic Editor

PLOS ONE

Additional Editor Comments (optional):

Authors have revised the manuscript to the standard which merit publication in Plos One. I accept the manuscript with a minor edit which authors should take care of before in the final manuscript: instead of using 'WH' they should use 'water hyacinth' throughout the manuscript.
---

## [Editor Report · Acceptance letter]

24 Aug 2020

PONE-D-20-11958R2 

Socioeconomic Effects of Water Hyacinth (Echornia Crassipes) in Lake Tana, North Western Ethiopia 

Dear Dr. Enyew:

I'm pleased to inform you that your manuscript has been deemed suitable for publication in PLOS ONE. Congratulations! Your manuscript is now with our production department. 

Kind regards, 

on behalf of

Dr. Ali Bajwa 

Academic Editor

PLOS ONE